# Phenotypic and Functional Characterization of Bovine Adipose-Derived Mesenchymal Stromal Cells

**DOI:** 10.3390/ani14091292

**Published:** 2024-04-25

**Authors:** Vitali V. Maldonado, Sriya Pokharel, Jeremy G. Powell, Rebekah M. Samsonraj

**Affiliations:** 1Department of Biomedical Engineering, University of Arkansas, Fayetteville, AR 72701, USA; vvm001@uark.edu (V.V.M.); sp072@uark.edu (S.P.); 2Department of Animal Science, University of Arkansas, Fayetteville, AR 72701, USA; jerpow@uark.edu; 3Interdisciplinary Graduate Program in Cell and Molecular Biology, University of Arkansas, Fayetteville, AR 72701, USA; 4Department of Orthopedic Surgery, University of Arkansas for Medical Sciences, Little Rock, AR 72205, USA

**Keywords:** bovine mesenchymal stem cells, phenotypic characterization, cellular therapy, regenerative medicine, cattle industry

## Abstract

**Simple Summary:**

In this study, we focused on obtaining mesenchymal stem cells (MSCs) from the fat tissue of cows and investigating their characteristics and functions. We collected fat tissue samples from healthy cows and used specific methods to isolate MSCs. We tested the cells’ ability to form colonies, grow and divide, express stem cell markers on the cell surface, differentiate into bone and fat cells, and produce a substance called indoleamine 2,3-dioxygenease (IDO) that helps regulate the immune system. The results showed successful isolation of MSCs from the cows’ fat tissue that could be grown and expanded in culture for an extended period. The MSCs also secreted significant amounts of IDO, indicating their potential to modulate the immune system and control inflammation. This study has important implications for the cattle industry, as it suggests that autologous (from the same individual) MSCs derived from fat tissue could be used as a complementary therapy for various diseases in cows. These MSCs may provide additional benefits compared to conventional treatments by addressing inflammation and tissue scarring associated with common cattle diseases. The methodology used in this study can be adopted by veterinary cell therapy labs to prepare MSCs for managing diseases in cattle, either from the same individual or from other donors.

**Abstract:**

Background: Mesenchymal stem cells (MSCs) are increasingly trialed in cellular therapy applications in humans. They can also be applied to treat a range of diseases in animals, particularly in cattle to combat inflammatory conditions and aging-associated degenerative disorders. We sought to demonstrate the feasibility of obtaining MSCs from adipose tissue and characterizing them using established assays. Methods: Bovine adipose MSCs (BvAdMSCs) were isolated using in-house optimized tissue digestion protocols and characterized by performing a colony formation assay, cell growth assessments, cell surface marker analysis by immunocytochemistry and flow cytometry, osteogenic and adipogenic differentiation, and secretion of indoleamine 2,3-dioxygenease (IDO). Results: Our results demonstrate the feasibility of successful MSC isolation and culture expansion from bovine adipose tissues with characteristic features of colony formation, in vitro multilineage differentiation into osteogenic and adipogenic lineages, and cell surface marker expression of CD105, CD73, CD90, CD44, and CD166 with negative expression of CD45. BvAdMSCs secreted significant amounts of IDO with or without interferon–gamma stimulation, indicating ability for immunomodulation. Conclusions: We report a viable approach to obtaining autologous adipose-derived MSCs that can be applied as potential adjuvant cell therapy for tissue repair and regeneration in cattle. Our methodology can be utilized by veterinary cell therapy labs for preparing MSCs for disease management in cattle.

## 1. Introduction

Among all domesticated species, cattle have major significance in the economics of the livestock industry, with 69.6 million tons of meat and 811 million tons of milk produced worldwide in 2017 [1,2]. There are several medical conditions that can negatively impact meat and milk production as well as reproductive efficiency in cattle. For cattle with high economic potential, these losses pose significant costs to the livestock industry. For example, bovine respiratory diseases are economically devastating in the US cattle industry costing producers over an estimated USD 500 million each year, and treatment has exclusively involved antibiotics [3]. There is heightened immune activity associated with respiratory diseases in cattle that could result in lung tissue destruction [4]. Other diseases that affect cattle include mastitis, digestive disorders, encephalopathies, and white muscle disease, to name a few [5]. Considering the potential of stem cells in tissue repair and regeneration, obtaining mesenchymal stem/stromal cells (MSCs) from bovine adipose tissue could be utilized for addressing a variety of disorders that require anti-inflammatory interventions and pro-regenerative functions.

MSCs are multipotent stem cells that are characterized by their ability to adhere to plastic, express cell surface markers CD105, CD73, and CD90, and lack surface markers CD45, CD34, CD14, or CD11b, as well as the ability to differentiate into osteoblasts, adipocytes, and chondroblasts [6]. In addition, MSCs have been shown to secrete growth factors and other soluble molecules such as interleukins, cytokines, and chemokines that play critical roles in tissue repair and regeneration. Owing to their self-renewal, regenerative and immunomodulation properties, MSCs function as medicinal signaling cells by virtue of their secretion of soluble factors and proteins that are anti-inflammatory, anti-fibrotic, anti-microbial, anti-apoptotic, and pro-regenerative [7]. MSCs inhibited *Staphylococcus aureus* biofilm formation in vitro and disrupted the growth of established biofilms [8]. MSCs have been trialed in treating bovine mastitis [9,10], and bone repair in small animal models [11]. In treating respiratory infections, the risk of antibiotic resistance may remain, however, it can be modulated with the adjunct use of MSCs, wherein the MSCs may serve to aid in the recovery of damaged tissue. MSCs also produce antimicrobial peptides (AMPs)—short peptides commonly found in neutrophils or epithelial cells [12]. AMPs kill bacteria directly by disrupting the integrity of the microbial membrane, or by inducing the release of proinflammatory cytokines and in turn the recruitment of immune cells. MSCs do not necessarily engraft or differentiate in vivo but secrete bioactive factors that stimulate the microenvironment to fight infection and restore host tissue. Additionally, human MSCs secrete indoleamine 2,3-dioxygenase (IDO) [13], which contributes to the immunosuppressive effects of these cells [14]. IDO converts tryptophan into N-formyl-kynurenine [15] that suppresses the activity of T effector cells and promotes regulatory T cell proliferation, preventing a potentially exaggerated immune reaction [16]. In addition, MSCs secrete soluble molecules, such as nitric oxide, prostaglandin E2 (PGE_2)_, IL-10 and transforming growth factor beta-1 (TGFβ1). The secretion of these factors suppresses the proliferation and/or activity of a variety of immune cells, including T cells, B cells, natural killer cells, and dendritic cells, as well as activated T_regs_ [13].In humans, MSC clinical trials have been successful in the treatment of graft-vs-host disease [17], amyotrophic lateral sclerosis [18], stroke [19], rheumatoid arthritis [20], and respiratory disorders including COVID-19-related complications [21]. 

Of the several tissue sources of MSCs, adipose tissue-derived MSCs are particularly advantageous owing to their ease of isolation and higher yield in cell numbers obtained from primary tissue digests compared to bone marrow tissues [22,23]. In cellular therapy applications, significant numbers of cells are required during transplantation or administration and, therefore, adipose tissues are regarded as a better source of MSCs. Furthermore, acquiring adipose tissue from animals is comparatively less invasive than bone marrow aspiration.

In this research article, we demonstrate the feasibility of isolating bovine MSCs from adipose tissue and characterizing the cells in vitro based on established phenotypic and functional assays [24,25]. MSCs were assessed for colony formation, self-renewing proliferative ability, presence of characteristic cell surface markers, differentiation capacities, and immunomodulatory features. Our preliminary findings in obtaining potent populations of MSCs highlight the viability of our approach, which holds significant potential for the development of a new generation of ‘living’ drugs for treating a range of diseases that may require anti-inflammatory and regenerative properties.

## 2. Materials and Methods

### 2.1. Collection of Adipose (Fat) Tissue from Cows

All procedures were performed in accordance with institutionally approved IACUC protocol #21092. After restraining a cow in a squeeze chute, the tailhead area (either side of the mid-line over the pin bones) was shaved of hair, disinfected with chlorhexidine scrub solution and isopropyl alcohol, then the incision site was injected with 10 mL of 2% lidocaine solution under the skin and into the adipose tissue. After sterilizing the site, a straight-line incision (approximately 1.5 to 2 inches) through the skin was made parallel to the spine using a sterile scalpel. Adipose tissue was removed using the scalpel, tissue scissors and sterile forceps. The incision area was closed with absorbable suture and tissue glue. The incision site was then cleaned, and an insect repellent was applied. Once retrieved from the animal, adipose tissue was washed extensively with sterile cold saline supplemented with 400 μg/mL streptomycin and 400 units/mL penicillin (cat.no. 15140-122, Gibco™, part of Thermo Fisher Scientific, Pittsburgh, PA, USA) prior to further tissue digestion.

### 2.2. Isolation of Bovine Adipose-Derived Mesenchymal Stem Cells (BvAdMSCs)

The adipose tissue sample was collected in PBS solution supplemented with 1% penicillin-streptomycin (P/S) (Gibco™) solution. After surface disinfecting the designated work area in a biosafety cabinet with 70% ethanol, the collected tissue sample was transferred to a 10 cm cell culture plate. The tissue sample was then minced carefully for about 1 h using a sterile scalpel blade (Figure 1). For the digestion of minced tissue, the tissue was placed in a tube containing Alpha Minimum Essential Medium (MEM α, cat.no. 12571-063, Gibco™) supplemented with 0.001% type I collagenase (cat.no. 17100017, Gibco™), 1% penicillin–streptomycin (P/S) and 50 µg/mL nystatin. The tissue was then incubated for digestion for 3 h. After completing the 3 h digestion, collagenase media was neutralized by adding equal volume of MEM α (Gibco™) supplemented with 15% Fetal Bovine Serum (FBS, cat.no. 26140-095, Gibco™), 50 µg/mL nystatin, 80 µg/mL amikacin sulfate and 1% penicillin–streptomycin (P/S). Then, digested tissue was centrifuged at 300 g for 10 min. The cell pellet was then resuspended in 20 mL of Alpha Minimum Essential Medium (MEM α, Gibco™) supplemented with 15% Fetal Bovine Serum (FBS, Gibco™), 50 µg/mL nystatin, 83.4 µg/mL amikacin sulfate and 1% penicillin–streptomycin (P/S) solution and was centrifuged again for 10 min at 300 g. After centrifugation, the supernatant was aspirated without disturbing the cell pellet, which was then resuspended in Alpha Minimum Essential Medium (MEM α, Gibco™) supplemented with 15% Fetal Bovine Serum (FBS, Gibco™), 50 µg/mL nystatin, 80 µg/mL amikacin sulfate and 1% penicillin–streptomycin (P/S) solution (hereafter referred to as ‘culture maintenance medium’). The cells were then plated in 10 cm dishes with the same culture maintenance medium (Figure 1). Plated cells were maintained in incubators at 37.5 °C with 5% CO_2_. The culture medium was changed 24 h after the initial seeding to remove any non-adherent cells from the culture. All in vitro cell cultures were maintained in humidified incubators at 37.5 °C with 5% CO_2_. 

### 2.3. Cell Culture and Maintenance

The culture medium was replaced every other day after the first media change that was performed 24 h post seeding. The cells were monitored regularly to assess their growth and the first colonies were observed within a 10–14-d period after the initial plating. At around 70–80% confluency, the cells were trypsinized (TrypLE™ Express, cat.no. 12605-010, Gibco^TM^) and replated in new 10 cm dishes at around 3000 cells/cm^2^. For this process, the cell plate was washed twice with phosphate buffered saline (PBS, pH 7.4, cat.no. 10010-023, Gibco™) and the cells were incubated with trypsin for 4 min. The trypsin activity was neutralized with double the volume of culture maintenance media, after which the cells were scraped carefully using a cell scraper. The cell suspension was centrifuged at 250 g/rcf for 5 min and the supernatant was aspirated without disturbing the cell pellet. The cell pellet was resuspended in culture maintenance media, and the cells were plated in new 10 cm dishes.

### 2.4. Morphological Characterization

Attachment of cells to the tissue culture polystyrene dishes and/or flasks were first observed at around 3–4 d post seeding of adipose tissue digests. The cells were monitored for the formation of circular colonies. This represents the ability of bovine adipose- derived mesenchymal stem cells to adhere to plastic as well as form colonies. The cells were monitored every day to check for the formation of new cell colonies as well as growth of existing cell colonies. The cells were monitored using an EVOS XL CORE imaging system at 4× magnification. The cells were split and passaged when multiple healthy colonies were observed in the flask.

### 2.5. Colony Forming Unit–Fibroblastic Assay

Colony forming unit–fibroblastic (CFU–F) ability was tested for the cells. The cells were divided into 2 groups with 150 cells per dish in one group (*n* = 3) and 500 cells per dish in the other group (*n* = 3). The cells were then maintained in culture for 2 week with a medium replacement on the 7th day. On the 14th day, the cells were washed twice with PBS and stained using crystal violet stain. The cell colonies were counted on each plate and the CFU–F ability was analyzed by plotting the average number of colonies formed against the number of cells seeded. By definition, an MSC colony should contain at least 50 cells when counted under microscopic examination.

### 2.6. Proliferation Assay

#### 2.6.1. Growth Assay

The growth of bovine adipose-derived mesenchymal stem cells was assessed over a 6-d period. On d 0, cells were seeded into eighteen 6 cm cell culture dishes at 3000 cells/cm^2^ in culture maintenance medium. Starting from d 1 until d 6, the cell count was obtained from 3 dishes each day (*n* = 3). To obtain the cell count, the cells were trypsinized for 4 min, after which the trypsin activity was neutralized by adding double the amount of culture maintenance medium. The cells were gently scraped, and a cell suspension was obtained. The cells were then counted using a hemacytometer and trypan blue staining. For a comparative understanding, human bone marrow-derived mesenchymal stem cells (hBM-MSCs) were also plated in an identical manner as the BvAdMSCs and the cell count was obtained for the human cells. 

#### 2.6.2. Cell Counting Kit-8 (CCK-8) Assay

The number of viable cells over time was determined similarly. BvAdMSCs were seeded into 12-well plates at a density of 3000 cells/cm^2^. Each day, starting from d 1 until d 6, the cells were trypsinized, neutralized with double the amount of culture maintenance medium, and scraped until the cells were detached from the flask. The cell suspension was collected in microcentrifuge tubes. Then, the cells were centrifuged at 1300 rpm for 5 min to obtain a cell pellet. The supernatant was discarded, and the cells were resuspended in 1 mL of cell culture maintenance medium. Cell suspension from each well (100 µL) was added to a 96-well plate in duplicates. The 96-well plate was incubated at 37 degrees Celsius and 5% CO_2_ for 6 h before adding the Cell Counting Kit-8 (cat.no. 96992, Millipore Sigma, Saint Louis, MO, USA) solution. The plate was then incubated for 2 additional hours. A plate reader was used to detect the absorbance at 450 nm. The cell number was determined using a standard curve made by known numbers of viable cells and their respective absorbances.

### 2.7. In-Vitro Osteogenic and Adipogenic Differentiation

The potential for bovine adipose-derived mesenchymal stem cells (BvAdMSC) to differentiate into osteogenic and adipogenic lineages was tested in vitro. For osteogenic differentiation, BvAdMSC cells were plated in a 6-well plate at 3000 cells/cm^2^. Twenty-four hours after the initial plating, the culture medium for the experimental group was replaced with alpha minimum media containing 15% FBS, 0.1 µM dexamethasone (cat.no. D4902, Sigma-Aldrich, Saint Louis, MO, USA), 50 µM ascorbic acid-2 phosphate (cat.no. A4544, Sigma-Aldrich), 10 mM β-glycerophosphate (cat.no. G9422, Sigma-Aldrich), 83.4 µg/mL amikacin sulfate and 1% penicillin–streptomycin solution. For the control group, a media change was performed using the regular culture maintenance medium. The cells were cultured in this medium for 28 d with a culture medium replacement every 2 d. For adipogenic differentiation, the cells were plated in a 6-well plate at 5000 cells/ cm^2^. Cells were maintained in culture for 2 d in the above mentioned culture conditions, after which the media for the experimental group was changed to high glucose DMEM (Dulbecco’s Modified Eagle Medium, cat.no. 11965092, Gibco^TM^) supplemented with 10% FBS, with or without 1 µM dexamethasone, 10 µM insulin (cat.no. SAFC-91077C, Sigma-Aldrich), 100 µM indomethacin (cat.no. I7378, Sigma-Aldrich), 11.5 µg/mL 3-isobutyl-1-methylxanthine (cat.no. I5879, Sigma-Aldrich), 83.4 µg/mL amikacin sulfate and 1% penicillin–streptomycin solution. The cells were maintained in the culture for 35 d with a medium replacement every 2 d.

To confirm whether the cells had differentiated into osteogenic and adipogenic lineages, they were stained using Alizarin Red staining and Oil Red O staining, respectively. For visualization of osteogenic differentiation, the cells were washed twice with PBS and fixed with 4% paraformaldehyde (PFA) for 10 min. The cells were washed again twice with PBS before adding 1% Alizarin Red Solution (pH 4.2). The cells were incubated at room temperature for 1 h before washing twice with PBS. To visualize the adipogenic differentiation, the cells were washed with PBS before fixing with 4% PFA for 1 h at room temperature. After 1 h, the cells were washed once with di-H_2_O before adding the 0.36% Oil Red O solution. The cells were incubated at room temperature for 1 h, after which they were washed twice with 60% isopropanol. The cells were then washed multiple times with water (H_2_O) to replace any background/precipitates. The dishes were left to dry inside a hood and the stained cells were visualized using a bright field microscope.

### 2.8. Surface Marker Characterization

#### 2.8.1. Microscopic Imaging

The surface marker characterization was performed for the BvAdMSCs using immunocytochemistry techniques. For the characterization of surface markers, cells were seeded at a density of 3000 cells/cm^2^ in 35 mm glass bottom dishes (cat.no. P35G-1.5-14-C, MatTek Life Sciences, Ashland, MA, USA). The cells were grown until 80% confluency after which they were fixed using 4% paraformaldehyde (PFA) solution. CD90 antibody conjugated with Alexa Fluor 488 (cat.no. NBP2-47755AF488, Novus Biologicals, Centennial, CO, USA) was used. For CD45 staining, purified CD45 antibody and Goat anti-Mouse IgG (H + L) secondary antibody (rhodamine) (Pre-adsorbed) (cat.no. NBP1-73133-1, Novus Biologicals) were used. Cells fixed in 4% PFA were permeabilized using 0.5% Triton-X 100, after which they were blocked in 5% horse serum. The cells were washed 3 times with PBS and incubated overnight in 15 µg/mL antibody prepared in 1% horse serum. After overnight incubation, the cells were washed in PBS and double stained for nucleus visualization using 5 µg/mL DAPI stain. The cells were then imaged using a fluorescence microscope to examine the expression of surface markers.

#### 2.8.2. Flow Cytometric Analysis

Flow cytometry was used to analyze the expression of CD44, CD45, CD73, CD90, CD105, and CD166. BvAdMSCs were seeded into T150 flasks and were grown until 70–80% confluence. The cells were then trypsinized and incubated for 5 min at 37 degrees Celsius, 5%CO_2_. Then, twice the amount of culture maintenance medium was added, and the flasks were gently scraped to obtain a cell suspension. The cell suspension was transferred to a tube and centrifuged at 1300 rpm for 5 min to obtain a cell pellet. The supernatant was discarded, 5 mL of PBS were added to the pellet and the solution was mixed. Cell counting was determined using an automatic cell counter (Countess, Invitrogen, part of Thermo Fisher Scientific, Pittsburgh, PA, USA). The cells were then centrifuged, and the supernatant was discarded. About 1 million BvAdMSCs were resuspended in 100 µL of FACS buffer (1% BSA, 0.1% sodium azide in PBS). The solution was mixed, and the cell suspension was then aliquoted into eight different microcentrifuge tubes. The tubes were labeled as their respective antibody. Then, CD44 (cat.no. NBP2-22530PE, Novus Biologicals), CD45 (cat.no. MA1-81458, Thermo Fisher Scientific, Carlsbad, CA, USA), CD73 (cat.no. LS-C723375-100, LSBio, Shirley, MA, USA), CD90 (cat.no. NBP2-47755PE, Novus Biologicals), CD105 (primary: cat.no. 10862-1-AP, Proteintech, Rosemont, IL, USA and secondary: cat.no. A-11008, Invitrogen), CD166 (cat.no. LS-C720014-100, LSBio), isotype control PE (cat.no. LS-C742375-0.1, BD Biosciences, Franklin Lakes, NJ, USA), and isotype control FITC (cat.no. LS-C149290-100, BD Biosciences) were added to the tubes following the manufacturer’s recommended amount. The cells were incubated at 4 degrees Celsius for 30 min, protected from light (the CD105 labeled tube was incubated for 1 h in primary antibody and 45 min in secondary antibody, with three FACS buffer washes in between incubation periods). Then, 500 µL of the prepared FACS buffer was added, and the mixture was centrifuged at 1500 rpm for 5 min. The supernatant was removed, and the cell pellet was washed twice more using 500 µL of FACS buffer. After the last wash, 300 µL of FACS buffer was added, the solution was mixed, and the cells were transferred to tubes suitable for flow cytometric analysis. The tubes were kept on ice and protected from light until the flow cytometry analysis was performed. The instrument used to analyze the samples was the MA900 Multi-Application Cell Sorter from Sony Biotechnology. The control isotype was gated at 5%. The data analysis and histogram plotting were performed using the FlowJo (version 10) software. 

### 2.9. Indoleamine 2,3-Dioxygenase (IDO) Activity Assay

The bovine adipose-derived MSCs were tested for secretion of IDO to assess immunosuppressive capabilities. The IDO assay was performed using an IDO Elisa kit (cat.no. MBS028821, MyBioSource, San Diego, CA, USA). BvAdMSCs were seeded in a 12-well plate at 5000 cells/cm^2^. Cells were plated as control and IFN-γ treatment groups. The treatment group cells were further divided into IFN-γ concentrations of 100 ng/mL and 200 ng/mL. The cells were grown to 75–80% confluency, after which the IFN-γ groups were treated with 100 ng/mL and 200 ng/mL concentrations. After 72 h, the conditioned media of cells treated with IFN-γ was collected for IDO assay and the cells were lysed using RIPA buffer to obtain a cell lysate. IDO assay was performed following the manufacturer’s instructions using the IDO Elisa kit (cat.no. MBS028821, MyBioSource).

### 2.10. Statistical Analyses

To compare the IDO secretion between stimulated and non-stimulated cells, and proliferative potential of MSCs from human bone marrow and bovine adipose tissues, *t*-test analyses were performed at each time point. A *p* value < 0.05 was considered to be significant.

## 3. Results

### 3.1. Bovine Adipose Tissue Yields Self-Renewing and Proliferative Stromal Cells

The tissue sample obtained from bovine adipose tissue was successfully cultured to isolate mesenchymal stem cells. Further characterization of isolated cells using morphological characteristics, differentiation abilities, and surface marker characterization confirmed the cells were mesenchymal stem cells. Bovine MSCs showed morphological characteristics of MSCs: the cultured bovine adipose-derived mesenchymal stem cells adhered to the plastic dishes after seeding and formed colonies at around 10–14 d after plating. The cells formed circular cell colonies with cells showing a fibroblastic phenotype (Figure 2). 

The bovine adipose-derived mesenchymal stem cells also exhibited colony forming units–fibroblastic (CFU–F) abilities. The group with the highest density of cells seeded (i.e., 500 cells per dish) showed a higher number of colonies formed compared to the group with lower density of cells seeded (i.e., 300 cells per dish) (Figure 3A,B). The colonies formed by BvAdMSCs were of variable sizes. Microscopic examination of stained colonies indicated that BvAdMSCs meet the criteria for definition as an MSC colony, comprising of at least 50 cells per colony (Figure 3A).

BvAdMSCs and hBM-MSCs were plated as described above to analyze their growth over a 6-d period. BvAdMSCs showed growth over the 6-d period, with a peak growth observed at d 5 post seeding. Comparatively, hBM-MSCs also showed growth over the 6-d period, with peak cell growth observed at d 4 post seeding (Figure 3C). Additionally, the number of viable cells increased over the 6-d post-seeding period, as indicated by the 450 nm absorbance after Cell Counting kit-8 solution addition (Figure 3C). All BvAdMSCs groups show consistent increase in light absorbance, correlating to a higher number of viable cells. Culture BvAdMSC 04 showed the highest increase in the number of viable cells while culture BvAdMSC 03 showed the lowest increase in the number of viable cells. These results show that BvAdMSCs have self-renewal ability without any inducing agents or exogenous supplementation of growth factors, and exhibit a proliferation rate similar to human MSCs. 

### 3.2. Bovine Adipose MSCs Express Characteristic Cell Surface Markers

The bovine adipose-derived mesenchymal stem cells were tested for the positive MSC surface markers CD105, CD73, CD90, CD44, and CD166 and negative MSC surface marker CD45. CD105 (Endoglin), CD73 (NT5E), and CD90 (Thy-1) are glycoproteins that are routinely used as positive MSC markers (as suggested by The Mesenchymal and Tissue Stem Cell Committee of the International Society for Cellular Therapy). CD44 is a transmembrane glycoprotein that is ubiquitously expressed in the body cells and CD166 (ALCAM) is a glycoprotein known for its role in cell adhesion [26,27]. High expression of CD105, CD73, CD90, CD166, and CD44 surface protein has been reported in undifferentiated human MSCs [28,29,30,31]. Flow cytometric analysis on BvAdMSC 02 and BvAdMSC 04 showed that both cell populations were positive for CD90 (96.3% and 99.4%), CD44 (99.3% and 99.7%), CD166 (81.2% and 67.3%), CD73 (79.6% and 63.1%), CD105 (55% and 26.4%), and negative for CD45 (3.34% and 5.23%), respectively, with the isotype control gated at 5% (Figure 4).

Additionally, we found that immunocytochemistry studies performed on the bovine adipose-derived MSCs showed positive expression for the CD90 surface marker. The CD90 surface marker was conjugated with Alexa 488 fluorophore, represented by a distinct green color on the stained cells (Figure 5A, left panel). The CD45 surface marker is a membrane glycoprotein with a molecular mass of about 200 kDa, that is a leukocyte common antigen and is expressed on most hematopoietic cells [32,33]. The cells were negative for the CD45 surface marker, which is characteristic to MSC populations (Figure 5A, right panel).

### 3.3. Bovine Adipose MSCs Differentiate into Osteo and Adipogenic Lineages

The multilineage differentiation ability of BvAdMSCs was assessed in vitro to verify their identify as MSCs as part of the minimal criteria to define a mesenchymal stem cell. The cells treated for osteogenic differentiation showed morphological changes within 2 week of treatment, as indicated by mineralization accumulated between cells and increased cell size. The cells were stained positively with Alizarin Red staining, that could be visualized both macroscopically and under a bright field microscope (Figure 5B) at 10× magnification. Under adipogenic induction, BvAdMSCs exhibited a rounded morphology, compared to the control group cell’s spindle like morphology, early in the culture. After 3–4 week of adipogenic induction, lipid droplets began to form. Upon staining with Oil Red O, the lipid droplets were noted to be stained bright red with visible clusters of droplets under examination by the light microscope (Figure 5C). 

### 3.4. Bovine Adipose MSCs Have Inducible IDO Activity and Immunosuppressive Potential

Indoleamine 2,3-dioxygenase is an immunosuppressant enzyme secreted by MSCs. IDO is a cytoplasmic protein responsible for converting tryptophan into N-formyl-kynurenine, which suppresses the proliferation of T-cells [15]. MSCs are routinely investigated for their functional features as a measure of their efficacy and potency. IDO secretion is indicative of MSC ability for immunomodulation. We assessed BvAdMSCs for secretion of IDO to evaluate their immunosuppressive abilities. At 70–80% confluency, BvAdMSCs were treated with IFN-γ. In BvAdMSC03, we noticed an increase in IDO secretion with 100 ng/mL of IFN-γ; however, the increases were not significant. We observed that BvAdMSCs, even without stimulation, secreted substantial amounts of IDO without any significant differences upon stimulation with IFN-γ (Figure 6). This implies that naïve adipose MSCs are capable of immunomodulation, even without external stimulus. Together, our results suggest the functional potency of BvAdMSCs in mediating immunomodulatory functions.

## 4. Discussion

Cell-based therapies for bovine diseases are a valuable therapeutic tool for a wide range of applications that require regenerative and anti-inflammatory treatment regimens [34]. In this preliminary research, we demonstrate the feasibility of successful isolation and characterization of bovine adipose tissue-derived MSCs in vitro. We report our in-house optimized protocol to acquire and generate functionally potent MSCs from adipose tissue through a minimally invasive tissue acquisition approach. One of the major limitations in cellular therapy is the generation of sufficient cell numbers required to promote regenerative or immunomodulatory effects upon transplantation. While bone marrow is a viable tissue source of MSCs, the cellular yield per gram of tissue is comparatively less than in adipose MSCs. According to a study by Nuvo Pain Management Center in 2019, an adipose tissue sample of the same size as a bone marrow sample had a higher volume of mesenchymal stem cells. This suggests that, as well as not being as invasive, adipose tissue removal requires a smaller sample size compared to bone marrow. In addition, obtaining bone marrow aspirates from animals is rather invasive and painful, requiring the use of high dose sedatives or anesthetics. These bottlenecks prompted us to investigate the feasibility of obtaining alternate tissue sources of MSCs, such as adipose tissue. In our study, the tailhead region of the animal was deemed a safe site for adipose tissue collection, eliminating the need for both general anesthetics and costly, invasive procedures.

Post-tissue harvest, rigorously established protocols developed from our laboratory permitted effective isolation of adipose MSCs from digested tissues. In cellular manufacturing, efficient isolation involves obtaining single-cell colonies of MSCs in culture within 7–10 d of plating. In our work, visible colonies were observed to develop as early as 3 d post-plating, demonstrating the efficacy of our collagenase-mediated digestion protocols. Cellular yield of 0.8 million primary cells per gram of adipose tissue at 14 d post-plating indicated the robust self-renewal and proliferative capacities of adherent MSC colonies. Both proliferation rates and cell numbers acquired were indicative of efficient culture expansion without the need for exogenous addition of growth factors. Adipose MSCs, in general, are documented to have higher proliferative capacities than their bone marrow counterparts [35,36]. This self-renewing ability of MSCs documented in vitro is a critical determinant in successful treatment outcomes in translational and veterinary MSC therapy trials. We have previously demonstrated that the growth capacity of MSCs is an important attribute in determining the in vivo potency of MSCs [24,25,37]. In an earlier subcutaneous transplantation study led by our team, human MSCs with higher growth capacity were shown to promote increased new bone formation compared to low growth capacity MSCs [25]. Therefore, the proliferation ability demonstrated by all BvAdMSC cultures in our study holds significance in translational and cellular therapy applications. 

In assessing the quality of the in-house isolated adipose MSCs in vitro, an important parameter that ensures multipotency is the ability to differentiate into mesodermal lineages. BvAdMSCs used in our study successfully differentiated into osteogenic and adipogenic lineages (Figure 5). Confirming successful differentiation ability in vitro is critical to ensure that the cells have passed the necessary minimum criteria for MSC definition and would be suitable candidates for in vivo tissue repair (post-transplantation) as well as for cell-based therapies, for example, in respiratory disease management in cattle that would require engraftment and differentiation to repair damaged lung tissue [13,25].

The surface marker characterization performed in this study revealed that the isolated BvAdMSCs are positive for CD90 (>96%), CD44 (>99%), CD166 (>67%), CD73 (>63%), and CD105 (>26%), and negative for CD45 (<6%) when the control isotype is gated at 5%. The decreased expression of the glycoproteins CD166, CD73, and CD105 when compared to hBM-MSCs can be attributed to the species and source differences. MSC surface marker expression patterns change across animal species [38]. In particular, expression of CD105 drastically decreases in MSCs derived from ovine and goat species, while DC166 expression is decreased in MSCs derived from mouse [29]. Additionally, bovine-derived MSCs surface marker expression can vary depending on the tissue source [39,40,41]. Mesenchymal stem cells derived from bovine fetal adipose tissue had a significantly higher expression of CD90 than CD73 and CD105 [39], suggesting that the surface marker expression pattern reported in this study is native to BvAdMSCs.

Given that the trophic milieu of MSCs are largely mediated by secreted proteins, MSCs may act as living drugs, exerting beneficial paracrine effects that are immunomodulatory and tissue-reparative, together providing an overall improvement in fighting infection as well as encouraging faster tissue healing. MSCs secrete soluble molecules such as prostaglandin E2 (PGE2), indoleamine 2, 3-dioxygenase (IDO), interleukin IL-10, and transforming growth factor-beta 1 (TGFβ1) that suppress proliferation and/or activity of T cells, B cells, dendritic cells, and activate regulatory T-cells (Tregs) [13]. One of the principal immunosuppression mechanisms of human MSCs is the production of IDO. IDO is involved in L-tryptophan catabolism, leading to its depletion in the surrounding microenvironment and accumulation of kynurenin, which then inhibits T cell activation, proliferation, and overall functional activity of T cells, dendritic cells, and natural killer cells, among other important effects [13,42,43]. MSC immunosuppressive effects and IDO activity have been shown in cells obtained from bone marrow as well as adipose tissues across human and non-human animal species [44,45]. Our work confirms the secretion of IDO by BvAdMSCs even in the absence of IFNg, strongly suggesting that the baseline immunomodulation levels of our culture-expanded cells are significant and potent to mediate immunosuppressive effects. IDO secretion from human MSCs without the need for stimulators like IFNγ has been previously reported [46,47]. In the vision for our ongoing extensions to apply the current methodology in animal health research, our innovative MSC intervention involves use of living cells that display immunomodulatory capacities, while also having the potential to enhance tissue recovery. 

## 5. Conclusions

Through cell-to-cell interaction and secretion of multiple bioactive molecules, MSCs have functional capacity to create a reparative environment and promote regeneration. Together, our data suggest that MSCs can be successfully isolated from bovine adipose tissue and cultured in the laboratory to generate sufficient cells with self-renewal, multi-lineage differentiation, and immunomodulatory capacities. Creativity lies in harnessing the soluble factors and proteins secreted by BvAdMSCs which, as we have thus demonstrated here, are easily obtainable in large quantities from the animal’s own body fat (autologous) and can also be applied as allogeneic therapies by following appropriate biobanking strategies. Clinical trials in human subjects are increasing, with promising results for treating musculoskeletal disorders such as osteoarthritis and bone fractures; nervous system conditions such as Alzheimer’s disease and spinal cord injuries; cardiovascular disorders such as myocardial infarction and cardiomyopathy; and respiratory disorders such as pulmonary hypertension, acute respiratory distress syndrome, COVID-19, and chronic obstructive pulmonary disease, to name a few (clinicaltrials.gov). Given these successes, there is no doubt that MSCs hold potential to ameliorate bovine diseases, and our study has now confirmed the feasibility of producing cellular preparations of potent stem cells in the laboratory, as well as validated stem cell phenotypic characteristics for use in farms and veterinary trials.

## Figures and Tables

**Figure 1 animals-14-01292-f001:**
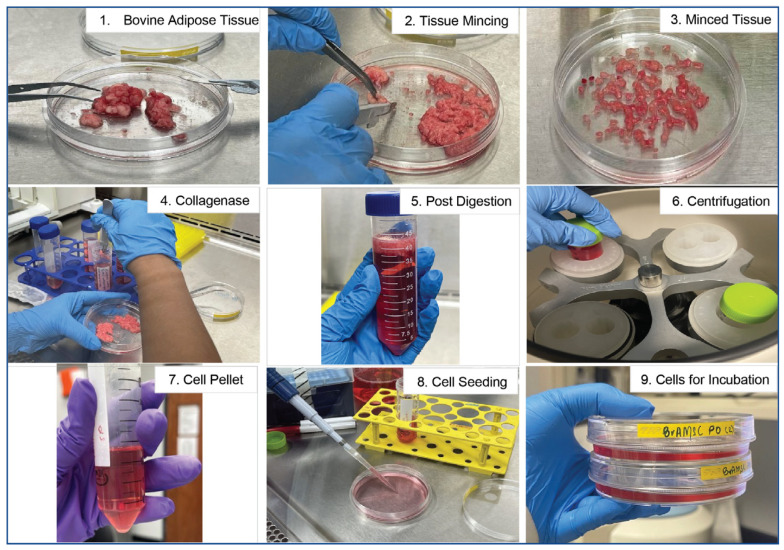
Schematic of bovine adipose tissue processing protocol. Stepwise illustration of the processing of bovine adipose tissue. Following mincing on petri dishes with a sterile scalpel (Steps 1–3), minced tissue is subjected to collagenase digestion (Steps 4–5) and centrifugation (Step 6). The cell pellet obtained post-centrifugation (Step 7) is resuspended in fresh complete media and seeded for culture expansion (Steps 8–9) then incubated for 1 week prior to formation of colonies of MSCs.

**Figure 2 animals-14-01292-f002:**
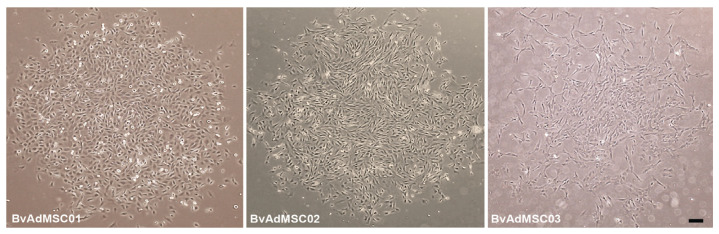
Bovine adipose tissue-derived MSC morphology. Phase-contrast images of bovine adipose tissue-derived MSCs from three individual animals (BvAdMSC 01, BvAdMSC 02, BvAdMSC 03) isolated successfully by plastic adherence. All bovine adipose tissue-derived MSCs adhered to plastic and were spindle-shaped in morphology. Colony formation was observed within 2 week of plating digested tissue fractions. Scale bar: 100 μm.

**Figure 3 animals-14-01292-f003:**
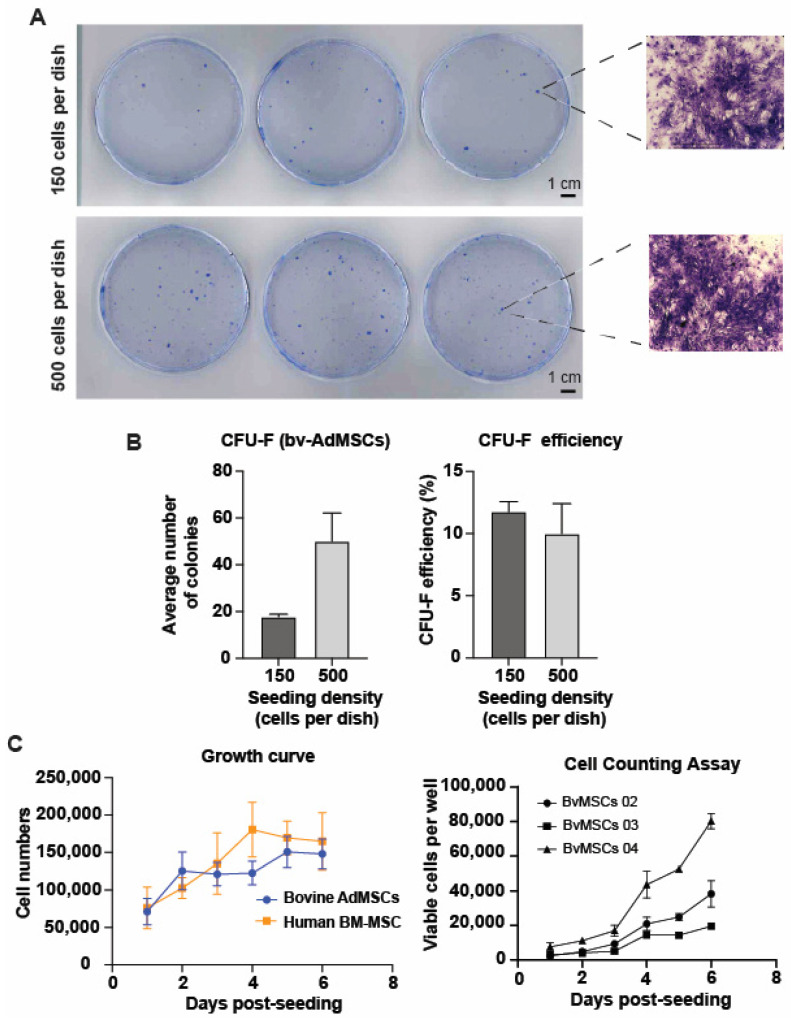
Phenotypic characterization: Colony formation and proliferative potential of bovine adipose tissue-derived MSCs. Adherent MSCs formed colonies when seeded at low densities, indicating their self-renewal ability and colony forming potential. (**A**). Scanned images of plates showing colonies of MSCs stained with crystal violet dye. Two different seeding numbers (150 cells per dish and 500 cells/dish) were tested. Scale bar: 1 cm. Microscopic examination of stained colonies at 4× magnification indicate that BvAdMSCs meet the criteria for definition as an MSC colony comprising of at least 50 cells per colony. (**B**). Bar graph showing the average number of colonies formed with two different seeding densities. The efficiency of colony formation (%) was similar between the two groups, together indicating that seeding densities did not affect overall CFU–F efficiency. (**C**). Line graph showing growth of BvAdMSCs in comparison with human bone marrow-derived MSCs. BvAdMSCs proliferated well in vitro over a period of 6 d, generating cell numbers similar to human MSCs. Cell counting kit assay using the Cell Counting Kit-8 solution to quantify the number of viable cells based on their absorbance at 450 nm. The line graph shows the viable cell number per well in a 12-well plate vs. days post-seeding from three different BvAdMSC groups (BvAdMSC 02, BvMSC 03, and BvMSC 04).

**Figure 4 animals-14-01292-f004:**
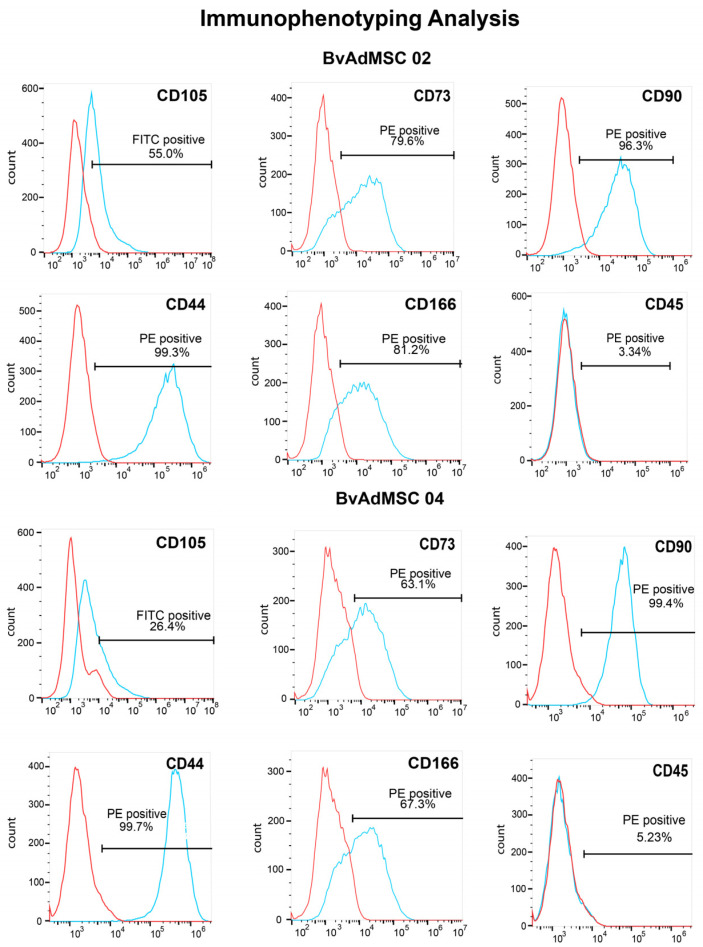
Flow Cytometric Analysis of BvAdMSCs Cell Surface Markers. Expression of typically positive mesenchymal stem cell markers CD105, CD73, CD90, CD44, and CD166 and typically negative mesenchymal stem cell marker CD45 were quantified using flow cytometry. The percentage of cells positive for a specific marker is indicated on the histogram. The surface marker histogram is denoted in blue, and the isotype control gated at 5% is denoted in red.

**Figure 5 animals-14-01292-f005:**
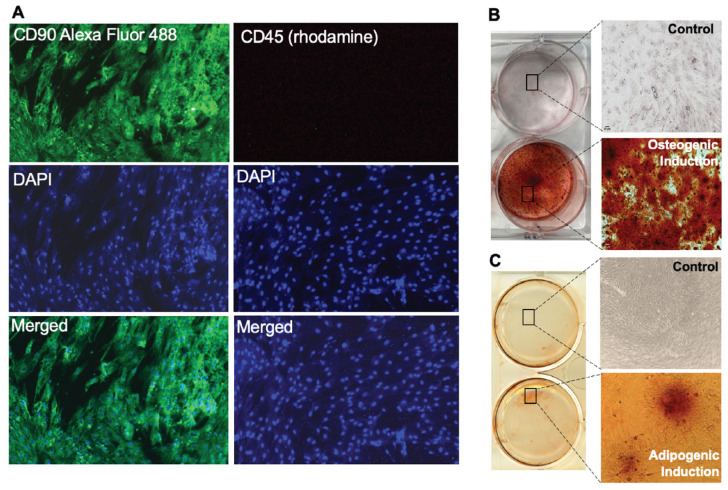
Phenotypic characterization: Surface marker expression and multilineage differentiation potential of bovine adipose-derived MSCs. (**A**). Immunocytochemistry performed on BvAdMSCs showed positive surface marker expression of CD90 (left panel) conjugated with Alexa Fluor 488 (green) and nuclear staining with DAPI (blue). Right panel showing negative expression of CD45 probed with corresponding secondary antibody conjugated with rhodamine (red). Lack of CD45 on BvAdMSCs is indicated by the absence of signal with immunocytochemical analysis. (**B**). BvAdMSCs subjected to osteogenic differentiation for 21 d and stained with Alizarin Red. Successful osteogenic differentiation is indicated by mineralization as seen by the red deposits when stained with Alizarin Red. (**C**). BvAdMSCs subjected to adipogenic differentiation for 21 d and stained with Oil Red O. Successful adipogenic differentiation is indicated by formation of lipid droplets as seen by the red deposits when stained with Oil Red O.

**Figure 6 animals-14-01292-f006:**
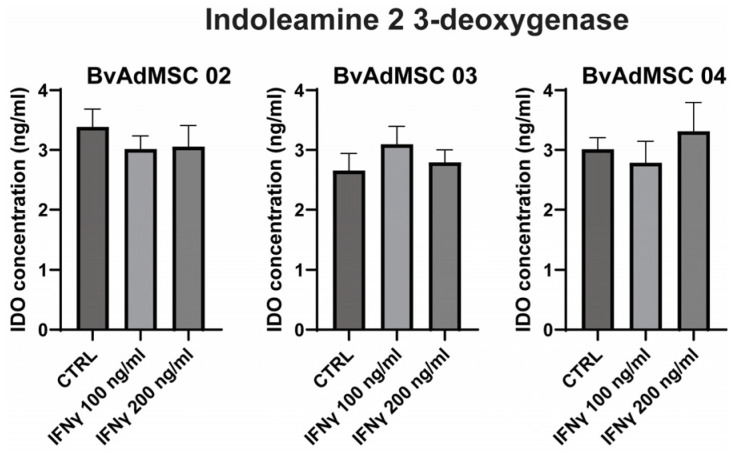
Functional characterization: Immunomodulation ability assessment by IDO secretion. Bar graphs showing ELISA-based assessment of the amounts of secretion of indoleamine dioxygenase by BvAdMSCs when stimulated with (100 ng/mL or 200 ng/mL) or without IFN-γ, indicating the ability of BvAdMSCs to act as immunosuppressive agents. Three different BvAdMSC groups (BvAdMSC 02, BvAdMSC 03, and BvAdMSC 04) were assessed. Statistical tests (*t*-tests) were performed between control and treatment groups; *p* values were greater than 0.05, indicating no significant differences between control and IFN-γ, together signifying that the baseline levels of IDO secretion were detectable even under non-stimulated conditions.

## Data Availability

The original contributions presented in the study are included in the article, further inquiries can be directed to the corresponding author.

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
