# Peer review of "Phenotypic and Functional Characterization of Bovine Adipose-Derived Mesenchymal Stromal Cells"

_animals, 2024, doi:10.3390/ani14091292_

Round 1
Reviewer 1 Report
Comments and Suggestions for Authors
I have reviewed the paper titled ‘Phenotypic and functional characterization of bovine adipose derived mesenchymal stromal cells’. It is very interesting study with very good findings. I advise the following comments and suggestions to be considered in their revision.
1/ While it is interesting that they characterize and in vitro expand the bovine derived mesenchymal stem cells into functional colonies, their claim of the potential of such cells to modulate the immune system and control inflammation is not strongly supported by sufficient data and should explained very well from an existed loiterature.
2/ To see the effect, degree of statistical association is required for the immunomodulation ability assessment by IDO secretion (Figure 6).
Author Response
Reviewer 1: Response to reviewer’s comments
I have reviewed the paper titled ‘Phenotypic and functional characterization of bovine adipose derived mesenchymal stromal cells’. It is very interesting study with very good findings. I advise the following comments and suggestions to be considered in their revision.
1. While it is interesting that they characterize and in vitro expand the bovine derived mesenchymal stem cells into functional colonies, their claim of the potential of such cells to modulate the immune system and control inflammation is not strongly supported by sufficient data and should explained very well from an existed literature.
Response: We thank the reviewer for the comment. We agree with the reviewer, and we have now included additional literature supporting the evidence of MSCs to modulate the immune system based on secretion of indoleamine 2,3 dioxygenase and other immunomodulatory functions mediated by interactions with T cells, B cells and NK cells. Please see page 3, lines 103-111, page 14, lines 523-529.
2. To see the effect, degree of statistical association is required for the immunomodulation ability assessment by IDO secretion (Figure 6).
Response: We thank the reviewer for the comment. We have performed the IDO assay by including two additional animal-derived BvAdMSCs to confirm our previous results. Statistical analysis was performed using t-tests between control and IFNγ treated groups. IDO expression did not differ significantly between the control and IFNγ treatment groups in all tested MSC populations (P>0.05). This suggests that MSCs secreted detectable baseline levels of IDO even without the addition of IFNγ, indicating their innate ability to modulate the immune system. Please see Figure 6 and page 12, lines 436-441.
Reviewer 2 Report
Comments and Suggestions for Authors
In this manuscript, Powell et al. describe the isolation, characterization, and extended culture of bovine-derived mesenchymal stem cells. First, they collected bovine adipose tissue from the tailhead area of a restrained cow, isolated cells from the tissue after homogenization, and cultured the cells in MEM-alpha with 15%FBS. After reseeding the cells, they subcultured and characterized the colonies via CFU-F and proliferation assays. They confirmed osteogenic and adipogenic differentiation potential and characterized some surface markers. Finally, they tested the MSC for immunosuppressive potential using IDO activity assay.
Overall, the manuscript is of interest, due to its translational potential for veterinary medicine. The methods are well described and the manuscript is clearly structured. However, it must be said that with a manuscript that short, the scientific content has to be watertight and I have some issues with the incomplete surface marker characterization.
Major:
The characterization of surface markers is essential when describing novel MSC isolates. I suggest that the authors perform a cytofluorimetric analysis for CD 44, CD45, CD90, CD105, CD 73, CD166, Oct34 and SSEA4.
Minor:
- Line 138: It says “After sterilization of the cell culture hood". Because you mostly didn't sterilize the cell culture hood, it would be better to describe it as "washed the work area within the biosafety cabinet with 70% Ethanol.
Comments on the Quality of English Language
Please carefully proofread the text for frequently missing articles. For example "Nonetheless, they can also be applied treat a range of diseases in animals.", or "After restraining cow in a squeeze chute"
Author Response
Reviewer 2: Response to reviewer’s comments
In this manuscript, Powell et al. describe the isolation, characterization, and extended culture of bovine- derived mesenchymal stem cells. First, they collected bovine adipose tissue from the tailhead area of a restrained cow, isolated cells from the tissue after homogenization, and cultured the cells in MEM-alpha with 15%FBS. After reseeding the cells, they subcultured and characterized the colonies via CFU-F and proliferation assays. They confirmed osteogenic and adipogenic differentiation potential and characterized some surface markers. Finally, they tested the MSC for immunosuppressive potential using IDO activity assay. Overall, the manuscript is of interest, due to its translational potential for veterinary medicine. The methods are well described and the manuscript is clearly structured. However, it must be said that with a manuscript that short, the scientific content has to be watertight and I have some issues with the incomplete surface marker characterization.
1. The characterization of surface markers is essential when describing novel MSC isolates. I suggest that the authors perform a cytofluorimetric analysis for CD 44, CD45, CD90, CD105, CD 73, CD166, Oct34 and SSEA4.
Response: We thank the reviewer for the comment. We agree that surface marker characterization is important in the context of this paper. Based on suggestions by the reviewer, we have now performed flow cytometric analysis on CD44, CD45, CD90, CD105, CD73, and CD166 using newly purchased antibodies as well as those previously available in the lab. Our results show expected expression for the above-listed antibodies. This new data is appended to the manuscript. Please see the new data provided in Figure 4. Also, please see corresponding texts in Page 1, lines 37-38, page 6-7, lines 281-310, page 10, lines 380-393, and page 14, lines 500-511.
2 Line 138: It says “After sterilization of the cell culture hood". Because you mostly didn't sterilize the cell culture hood, it would be better to describe it as "washed the work area within the biosafety cabinet with 70% Ethanol.
Response: We thank the reviewer for the comment. We have now rephrased the sentence following the reviewer’s suggestion to include the suggested wording with 70% ethanol. Please see page 3, line 146-147.
3. Please carefully proofread the text for frequently missing articles. For example "Nonetheless, they can also be applied treat a range of diseases in animals.", or "After restraining cow in a squeeze chute"
Response: We thank the reviewer for the comment. We have proofread the paper and fixed grammatical errors. Please see page 1, line 28; page 2, line 76; page 3, line 129; page 5, line 201, 214, and 216; page 6, line 260; page 8, lines 337 and 338; page 11, line 416; page 13, line 455.
Reviewer 3 Report
Comments and Suggestions for Authors
In this manuscript, Jeremy G. Powell and associates seek to explore the practicality of harvesting Mesenchymal Stem Cells (MSCs) from adipose tissue and their characterization via standard assays. The team successfully isolated MSCs from bovine adipose tissue (bv-AdMSCs) and confirmed their capabilities for sustained in vitro expansion, colony formation, and differentiation into osteogenic and adipogenic lineages. Notably, bv-AdMSCs secreted considerable levels of indoleamine 2,3-dioxygenase (IDO), both with and without interferon-gamma induction, showcasing their potential for immunomodulation. In conclusion, this study presents a feasible method for acquiring autologous MSCs that could enhance existing treatments for various inflammatory conditions and tissue scarring prevalent in cattle. However, there are critical issues that need addressing, as detailed below:
1. The abstract requires restructuring to succinctly highlight the central findings, as it currently provides insufficient detail on the results.
2. The arrangement of figures needs revision; all figures should be placed after the main body of text rather than being grouped together.
3. Typically, the introduction should not contain figures. Furthermore, Figure 1 suggests that bovine respiratory diseases are multifaceted, raising the question of how the authors can substantiate that only interleukin 6 (IL-6), tumor necrosis factor-alpha (TNF-α), interleukin 8 (IL-8), interleukin 17 (IL-17), interferons, and Toll-like receptors are the critical elements.
4. The results from the colony formation assay are suboptimal, with colonies appearing too small. If colonies remain diminutive after two weeks of culture, it implies that this assay may not be an appropriate measure of cell proliferation.
5. Beyond colony formation and cell counting, it is recommended to perform additional proliferation assays, such as the Cell Counting Kit-8 (CCK-8), to provide a more comprehensive evaluation of cell growth.
Comments on the Quality of English LanguageMinor editing of English language required
Author Response
Reviewer 3: Response to reviewer’s comments
In this manuscript, Jeremy G. Powell and associates seek to explore the practicality of harvesting Mesenchymal Stem Cells (MSCs) from adipose tissue and their characterization via standard assays. The team successfully isolated MSCs from bovine adipose tissue (bv-AdMSCs) and confirmed their capabilities for sustained in vitro expansion, colony formation, and differentiation into osteogenic and adipogenic lineages. Notably, bv-AdMSCs secreted considerable levels of indoleamine 2,3-dioxygenase (IDO), both with and without interferon-gamma induction, showcasing their potential for immunomodulation. In conclusion, this study presents a feasible method for acquiring autologous MSCs that could enhance existing treatments for various inflammatory conditions and tissue scarring prevalent in cattle. However, there are critical issues that need addressing, as detailed below:
1. The abstract requires restructuring to succinctly highlight the central findings, as it currently provides insufficient detail on the results.
Response: We thank the reviewer for the comment. The abstract has been revised to include data from additional experiments and restructured to highlight the central findings of the study. Please see Page 1, lines 35-43.
2. The arrangement of figures needs revision; all figures should be placed after the main body of text rather than being grouped together.
Response: We thank the reviewer for the comment. The figure placement is now revised, and the figures have been placed after the main body of text rather than being grouped together. Please see page 4, figure 1; page 8, figure 2; page 9, figure 3; page 11, figure 4; page 12, figure 5; and page 13, figure 6.
3. Typically, the introduction should not contain figures. Furthermore, Figure 1 suggests that bovine respiratory diseases are multifaceted, raising the question of how the authors can substantiate that only interleukin 6 (IL-6), tumor necrosis factor-alpha (TNF-α), interleukin 8 (IL-8), interleukin 17 (IL- 17), interferons, and Toll-like receptors are the critical elements.
Response: We thank the reviewer for the comment. We agree with the reviewer that figures are not typically part of the Introduction section. Therefore, we have removed the original Figure 1 that contained information on factors upregulated in bovine respiratory diseases. We also acknowledge that several other factors can go into play and have revised the manuscript to keep the theme generic without specific relevance to bovine respiratory disorders.
4. The results from the colony formation assay are suboptimal, with colonies appearing too small. If colonies remain diminutive after two weeks of culture, it implies that this assay may not be an appropriate measure of cell proliferation.
Response: We thank the reviewer for the comment. Our results from the CFU-F assay have shown evidence of colony formation when stained with crystal violet dye. By definition, an MSC colony should contain at least 50 cells. All visible colonies were examined under the microscope, and we can conclude that each colony contained more than 50 cells. The microscopic magnification is now included in the Figure 3A. Please see revisions to the section in Page 5, Lines 202-205.
5. Beyond colony formation and cell counting, it is recommended to perform additional proliferation assays, such as the Cell Counting Kit-8 (CCK-8), to provide a more comprehensive evaluation of cell growth.
Response: We thank the reviewer for the comment. We agree that additional proliferation assays were needed to assess BvAdMSCs growth. We have included additional data derived from the Cell Counting Kit- 8 (CCK-8) assay to the paper. Please see Figure 3C. Page 5, lines 219-233, and page 10, lines 371-376.
6. Minor editing of English language required
Response: We thank the reviewer for the comment. We have proofread the paper and fixed grammatical errors.
Round 2
Reviewer 2 Report
Comments and Suggestions for Authors
The authors have addressed the concerns raised during the previous review. They have performed additional experiments to characterize the stem cells used in this experiment, and carefully revised the manuscript.
I recommend publication.
Reviewer 3 Report
Comments and Suggestions for Authors
No more questions.